# MindGames: Targeting Theory of Mind in Large Language Models with Dynamic Epistemic Modal Logic

**Damien Sileo**
Univ. Lille, Inria, CNRS
Centrale Lille, UMR 9189
CRIStAL, F-59000 Lille, France
`damien.sileo@inria.fr`

**Antoine Lernould**
Univ. Lille
CRIStAL, F-59000 Lille, France

## Abstract

Theory of Mind (ToM) is a critical component of intelligence but its assessment remains the subject of heated debates. Prior research applied human ToM assessments to natural language processing models using either human-created standardized tests or rule-based templates. However, these methods primarily focus on simplistic reasoning and require further validation. Here, we leverage dynamic epistemic logic to isolate a particular component of ToM and to generate controlled problems. We also introduce new verbalization techniques to express these problems in English natural language. Our findings indicate that some language model scaling (from 70M to 6B and 350M to 174B) does not consistently yield results better than random chance. While GPT-4 demonstrates superior epistemic reasoning capabilities, there is still room for improvement. Our code and datasets are publicly available[1]

## 1 Introduction

Theory of Mind (ToM) is the cognitive ability to attribute mental states, such as beliefs, desires, and intentions, to oneself and others, allowing individuals to understand and predict behavior based on these inferred mental states. It is an important requirement for general text understanding or artificial intelligence (Navarro et al., 2020), but claims about ToM are prone to bias from human expectations (de Waal, 2016). Kosinski (2023) recently sparked debate by showing that scaling large language models (LLMs) improves performance at standardized tests designed to measure ToM. However, these tests were widely discussed in academic research and might have leaked into the training corpora of LLM. Earlier work generated synthetic examples instead, extending the bAbi (Weston et al., 2016) framework. Nematzadeh et al. (2018) proposed a dataset of fixed templates based on the

*Sally-Anne* problem (Baron-Cohen et al., 1985):
> *Sally puts a marble in a box while Anne is with her. Sally leaves for a moment and Mary puts the marble in a basket. Where will Sally look for the marble?* [ANSWER=BOX]

Le et al. (2019) deem these problems simplistic and extend them to track second-order beliefs (e.g. the belief of Sally about Anne's beliefs).

In our study, we generate dynamic epistemic logic (DEL) problems and develop verbalizations to transform them into natural language inference problems. DEL is a branch of modal logic that can model an individual's knowledge about particular facts or about other agents' knowledge. DEL also enables reasoning about the impact of consecutive public announcements:
> *Alice and Bob have mud on their head. Their father says that at least one of them is muddy. He asks Alice and Bob if they are muddy. Do Alice and Bob know that they are muddy?* [ANSWER=NO] *They answer that they don't know. Do Alice and Bob now know that they are muddy?* [ANSWER=YES]

Bob would have answered YES to the first question if Alice was not muddy, so after Bob's first answer, Alice can know that she is muddy.[2] DEL can formalize certain ToM problems, making it a valuable perspective for ToM assessment. The problems we create can require tracking multiple agents' beliefs and reasoning about higher-order beliefs[3]. Our dataset encompasses numerous variations of the *Muddy Children* and *Drinking Logicians* problems (van Eijck, 2014). This controlled test bench offers new appreciations of language model scaling and presents the first dataset with a complexity that can challenge supervised learning models. The dataset and the scripts to generate is publicly available[1].

---

[1][code:GitHub][data:HF-datasets]

[2]The same holds if we switch Bob and Alice.

[3]For example, Anne's belief about Sally's belief about Anne's belief about Mary's belief.

## 2 Related Work

**Logical Reasoning in Natural Language Processing** Logic shares profound connections with NLP. Early systems were built around logic, and more recent approaches incorporate logical reasoning into neural networks (Hamilton et al., 2022; Helwe et al., 2022). Another line of research closer to ours investigates the logical capabilities of NLP models using textual datasets and labels generated with logical reasoning tools. RuleTaker (Clark et al., 2020) explores this area with propositional logic, while LogicNLI addresses first-order logic (Tian et al., 2021). Richardson and Sabharwal (2022) examine the satisfiability problem in natural language. Sileo and Moens (2022) targets probabilistic logic. Our study is the first to focus on modal logic, specifically epistemic logic, in natural language.

**Theory of Mind in NLP** To measure ToM capabilities of NLP models, Nematzadeh et al. (2018) created examples using Sally-Ann templates, and Le et al. (2019) added complexity to the data by incorporating second-order knowledge. Both studies framed their examples as question-answering tasks. Kosinski (2023) employed handcrafted tests to evaluate language models' next-word prediction capabilities. Ullman (2023) showed LLM brittleness to interventions on these datasets and Ma et al. (2023) consolidated the prior datasets into a principled evaluation suite. The Social-IQA dataset (Sap et al., 2019) covers a broad spectrum of social commonsense, encompassing aspects of theory of mind and challenges like comprehending desires and emotions. Cohen (2021) investigated whether natural language inference models captured veridicality with epistemic verbs like *know* and *think*, using handcrafted patterns. This task was incorporated into the BIG-Bench framework (Srivastava et al., 2022) as the *epistemic-reasoning* task, but it targets only one shallow aspect of epistemic reasoning. Bara et al. (2021) used a Minecraft dataset for real-time belief deduction in collaborative tasks. Shapira et al. (2023b) highlighted LLM struggles in faux pas tests. Shapira et al. (2023a) conducted stress tests on LLMs' social reasoning capabilities.

**Epistemic Logic and ToM** Bolander (2018) showed that the Sally-Ann problem could be modeled with epistemic logic. Van Ditmarsch and Labuschagne (2007) examined more general connections between DEL and ToM, while Dissing and Bolander (2020) demonstrated DEL's applicability in robotics. Van De Pol et al. (2018) explored the plausibility of epistemic logic for ToM by investigating its theoretical computational tractability.

## 3 Dynamic Epistemic Logic Problem Generation and Verbalization

### 3.1 Problem definition

Our objective is to simultaneously create dynamic epistemic logic problems and their corresponding natural language representations, with a (PREMISE, HYPOTHESIS, LABEL) format.

An epistemic logic problem can be decomposed into the following components:

**Agents:** A set of $N$ individuals, each assigned a different arbitrary name.

**Predicates:** A set of Boolean predicates. Here, we use $N$ predicates, one corresponding to each agent (e.g., *Alice has mud on her head*).

**Observabilities:** The description of each agent's initial knowledge of the predicate values. We represent observabilities with a boolean matrix $\mathcal{O}$ of size $N \times N$, where $\mathcal{O}_{i,j}=1$ means that agent $i$ initially knows whether predicate $j$ is true.

**Announcements:** A list of expressions (predicates or agent knowledge about predicates) that are shared to all agents. Announcements are made sequentially, and each new announcement can change what the agents know, even if it is the same announcement is repeated twice.

**Hypothesis:** An expression that may contain predicates and knowledge of agents about particular expressions after the announcements, given the agents, observabilities, and announcements grouped into a premise.

### 3.2 Setups: connecting predicate and observabilities

The concrete choice of predicates dictates the structure of observabilities. For example, the predicate *"Alice has mud on her head"* is observable by agents other than Alice, but *"Alice has mud on her hand"* could be observable by everyone. We group predicates and observabilities into what we call *setups* to generate textual descriptions. We define the following setups:

**Forehead-mud setup**

PREDICATE$_i$: <AGENT$_i$>*'s forehead is muddy.*
$\mathcal{O}$ : ONES$(N) -$ IDENTITY$(N)$

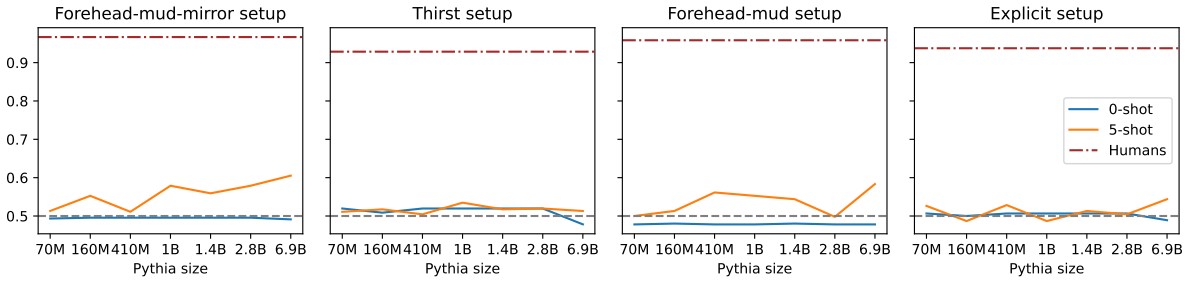

Figure 1: Accuracy of Pythia language models on MindGames setups.

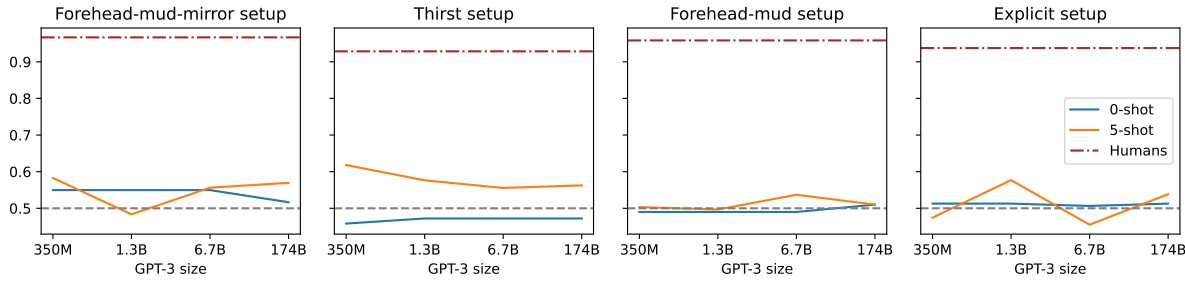

Figure 2: Accuracy of GPT-3 family (ada, cabbage, curie, davinci) language models on MindGames setups.

**Forehead-mud-mirror setup**

PREDICATE$_i$: <AGENT$_i$>'s forehead is muddy.
$\mathcal{O}$ : ONES($N$)
OBSERVATION: *There is a mirror in the room.*

**Thirst setup**

PREDICATE$_i$: <AGENT$_i$>'s is thirsty.
$\mathcal{O}$ : IDENTITY($N$)

**Explicit setup**

PREDICATE$_i$: <AGENT$_i$> picked a red card.
$\mathcal{O}$ : RANDBOOL($N, N$), $\mathbb{E}(sum(\mathcal{O}))=N$
OBSERVATION: *Each person draws a card, face unrevealed (red or black).* < <AGENT$_j$> *card is revealed to* <AGENT$_i$>. *for all* $i, j$ *where* $\mathcal{O}_{i,j}=1$>

### 3.3 Problem verbalization

We then construct a problem for a given setup with the following natural language template:

**[Premise]** *There are <N> persons. Everyone is visible to others.* <OBSERVATION> *It is publicly announced that someone* <PREDICATE> <[0 − N] ANNOUNCEMENTS>

**[Hypothesis]** <$[1 − K]^{th}$ ORDER BELIEF>

$[0 − N]$ denotes uniform sampling from 0 to $N$. We restrict announcements to first-order beliefs. A first-order belief has the following structure: <AGENT> (*can know whether* | *can know that* | *cannot know that* | *cannot know whether*)

(<PREDICATE>|<NEGATED-PREDICATE>), e.g. *Alice cannot know whether Bob is not muddy.* We use *can* to acknowledge that an agent could theoretically infer something but fail to see it. A $K^{th}$ order belief is a first-order belief about a $(K−1)^{th}$ order belief. We consider *everyone*, *not everyone*, and *nobody* as possible subjects for the setup predicates. Subjects are uniformly sampled among these quantifiers and the list of individual agents. We transform abstract problem representations into natural language and code that can be fed to a model checker to determine whether a hypothesis is entailed by the premise. We use the SMCDEL model checker (Benthem et al., 2018), an announcement logic based on the S5 (Lewis et al., 1959) modal logic. This implementation is the most cited publicly available epistemic logic as of April 2023. We discard examples where the premise contains a contradiction[4]. To generate diverse and gender-balanced random English surnames, we use CensusName[5] (Qian et al., 2022).

## 4 Experiments

### 4.1 Problem generation parameters

We randomly sample $N \in \{2, 3, 4\}$ agents, as we observed that problems were sufficiently challeng-

---

[4]We identify contradictions by examining whether an unused predicate is entailed or not by the premise.
[5]https://pypi.org/project/censusname/

ing with only three agents, and we use $K=2$ for the same reason. We use knowledge predicate negations 80% of the time to encourage richer inferences (as the fact that an agent does not know something conveys information to others) in announcements and 50% of the time otherwise.

## 4.2 Controlling for example difficulty

Shortcuts, like hypothesis only bias (Gururangan et al., 2018; Zhang et al., 2023), can lead to the answer without correct reasoning. To control for shortcuts, we trained a relatively shallow supervised model (deberta-small (He et al., 2021), 6 layers, 44M backbone parameters) on a training set combining all setups (ensuring that there was no duplicate and no example that was also in the test set). We used 11.2k training examples for 3 epochs and a learning rate of 3e-5 and 3.73k test and validation examples. Overall validation accuracy was 83%. We also experimented with simpler lexical baselines like TF-IDF which did not capture negations well enough. We assumed that examples correctly predicted by deberta-small with high confidence contained shortcut cues. We used these deberta-small predictions and confidence as additional metadata. We found that the evaluated language models already failed on easy examples. So we used a random subset of the validation and test subsets for our experiments, but our dataset can be filtered by difficulty using the provided confidence level and the discrepancy between deberta-small prediction and ground truth.

We limit the number of agents to 3 and deduplicate then undersample the problems to generate 400 test cases with a perfect balance of True/False labels per setup. We refer to the resulting dataset as MindGames.

## 4.3 Scaling experiments

We conduct zero-shot experiments and few-shots with a range of language models. We use standard prompting to follow Kosinski (2023) setup. We use the lm-eval-harness software (Gao et al., 2021) to measure whether a language model perplexity favors the correct reasoning in a multiple-choice setting, with a natural language inference prompt from Brown et al. (2020): <PREMISE> *Question:* <HYPOTHESIS> *True or False ?"* with two possible continuation choices, *True* and *False*. We evaluate two families of language models:

**Human evaluation**   We present 50 test samples per setup to two NLP researchers only instructed to perform entailment detection. Inter-annotator agreement is 0.89, and average accuracy is 94%[6].

**Pythia language models**   We select the Pythia (Biderman et al., 2023) language models for our open-source scaling experiments. We use the checkpoints trained on the deduplicated corpus (deduped) with checkpoint sizes of 70M, 160M, 410M, 1B, 1.4B, 2.8B, and 6.9B.

**OpenAI API**   We evaluate the OpenAI GPT-3 (Brown et al., 2020) models, specifically the ada, babbage, curie, and davinci checkpoints, through the public API. We assume that their model sizes are respectively 350M, 1.3B, 6.7B, and 174B and we use the default temperature.

Figure 1 displays the results for various Pythia model sizes. We observe that scaling improves 5-shot[7] reasoning, but it has no impact on zero-shot reasoning. In contrast to the emergence results reported by Kosinski (2023), Figure 2 does not show a clear scaling trend for GPT-3 models on MindGames data, which suggests that the emergent behavior they observed was not due to robust epistemic logic capabilities.

## 4.4 Qualitative analysis with ChatGPT

We also run brief qualitative analyses with GPT-3.5 and GPT-4 (OpenAI, 2023), as of May 2023. On 20 randomly sampled problems, we found that GPT3 was 60% correct and GPT-4 70% correct. We present a brief qualitative analysis of the respective models.

As shown in Appendix A, GPT3.5 tends to answer that there is not enough information and to perform correct inferences only when it requires very shallow reasoning. GPT-4 can solve this particular example. However, some problems are still challenging, as shown in Figure 3. GPT-4 rarely answers that there is not enough information and its reasoning looks has the surface form of epistemic reasoning, but occasionally contains glaring mistakes.

## 5 Conclusion

We developed a novel dataset aimed at evaluating epistemic logic reasoning, addressing a particular

---

[6]Most errors arose from failing to distinguish between *know whether* and *know that*.
[7]Increasing number of examples did not improve validation accuracy.

> **User:**
> [Premise] There are two persons. Everyone is visible to others. There is a mirror in the room. It is publicly announced that someone's forehead is muddy. It is publicly announced that not everyone's forehead is muddy. It is publicly announced that not everyone's forehead is muddy. [Hypothesis] Robert can now know whether or not everyone's forehead is muddy. Is the hypothesis true or false ?

> **GPT-4:**
> False. Based on the information provided, Robert cannot know for certain whether or not everyone's forehead is muddy. The statements provided are: There are two persons. Everyone is visible to others. There is a mirror in the room. It is publicly announced that someone's forehead is muddy. It is publicly announced that not everyone's forehead is muddy. Since it is publicly announced that not everyone's forehead is muddy, Robert can deduce that at least one person does not have a muddy forehead. However, this information alone does not allow Robert to definitively determine whether or not his own forehead is muddy.

Figure 3: GPT4 conversation

aspect of ToM. Our results reveal that this task continues to pose challenges for contemporary large-scale language models. When future models can solve MindGames for 2-3 agents, the difficulty can be easily scaled up with more agents. Future studies could better explore human performance on our dataset, taking into account factors such as age and educational background. Additionally, further investigation can examine the impact of fine-tuning on other downstream tasks and assess how well Transformer circuits model Kripke structures that represent modal logic problems.

## 6 Limitations

Theory of mind is a complex subject, and our study takes a deliberately specific angle, leaving multiple open problems:

**Language** Our work is centered on English, the method could be adapted to other languages using a subject-verb-object structure. Besides, we restricted our study to templates that do not cover the full variety of the English language.

**Prompt structure and models scaling** We focused on zero-shot and few-shot prompting, which were sufficient to (Kosinski, 2023), however,

Moghaddam and Honey (2023) recently showed that more advanced prompting schemes made significant differences. In addition, we did not explore the full range of Pythia models due to computational limitations.

**Task complexity, annotators variation** The task we proposed is relatively complex, and raises questions about the profiles of annotators that would match the results of a symbolic reasoner. The framework of DEL itself can also provide insights on theory of mind, as a DEL solver perfectly solves this task, even though we could feel uncomfortable attributing ToM to the solver. We might argue that failing on simple DEL examples disproves ToM, but proving failure is difficult, as mentioned in the previous paragraph.

## 7 Ethical considerations

This work involves human annotations. However, we used procedurally generated data, ensuring no confidential or harmful content. Besides, annotations were carried out during the researchers' working hours. For these reasons, our Institutional Review Board has determined that it was exempted from formal review according to internal guidelines.

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

## A    Qualitative example with GPT3.5

> **User:**
> [Premise] There are three persons. Everyone is visible to others. It is publicly announced that someone's forehead is muddy. It is publicly announced that Herbert knows whether someone's forehead is muddy. [Hypothesis] Herbert can now know that Herbert's forehead is muddy. Is the hypothesis true or false ?

> **GPT-3.5:**
> It is not possible to determine the truth value of the hypothesis based on the given premise. While it is publicly announced that someone's forehead is muddy, it is not clear who the muddy forehead belongs to. Therefore, it is not possible to determine whether Herbert's forehead is muddy or not based on this information alone. However, it is stated that Herbert knows whether someone's forehead is muddy, so Herbert may know whether their own forehead is muddy or not based on additional information not provided in the premise.

Figure 4: GPT-3.5 conversation