# OpenReview forum: "MindGames: Targeting Theory of Mind in Large Language Models with Dynamic Epistemic Modal Logic"
_EMNLP/2023/Conference — EMNLP 2023 Findings_

### Official Review · Reviewer_6Ahj · 2023-08-01

**Paper Topic And Main Contributions:** The paper offers a systematic way to …
**Soundness:** 4

**Excitement:**

3: Ambivalent: It has merits (e.g., it reports state-of-the-art results, the idea is nice), but there are key weaknesses (e.g., it describes incremental work), and it can significantly benefit from another round of revision. However, I won't object to accepting it if my co-reviewers champion it.

**Reasons To Accept:**

The systematic set of problems is interesting and well-designed.
The results are very clear: LLMs fail at the task, way below what humans do.

**Reasons To Reject:**

The failure of the models is not understood: is it due to some predicates which are not understood by the model in the same way as it is implemented in the logic? Is it because reasoning steps are hard to follow (a baseline control of inferences outside of the ToM domain may be feasible)?

**Reproducibility:**

5: Could easily reproduce the results.

**Reviewer Confidence:**

3: Pretty sure, but there's a chance I missed something. Although I have a good feel for this area in general, I did not carefully check the paper's details, e.g., the math, experimental design, or novelty.

---

> ### Author Rebuttal · Authors · 2023-08-28
>
> We sincerely thank you for the thoughtful comments and valuable feedback on our submission.
> Our qualitative analysis using ChatGPT indicates that while the problem statement is correctly understood, performing the reasoning itself remains difficult. This aligns with findings from the BIGBench results [1], which showed logic tasks like logical_deduction and analytic_entailment to be challenging for GPT-3.
>
> [1] Srivastava, A., Rastogi, A., Rao, A., Shoeb, A.A.M., Abid, A., Fisch, A., Brown, A.R., Santoro, A., Gupta, A., Garriga-Alonso, A. and Kluska, A., 2022. Beyond the imitation game: Quantifying and extrapolating the capabilities of language models. arXiv preprint arXiv:2206.04615.

---

### Official Review · Reviewer_fwC4 · 2023-08-02

**Typos Grammar Style And Presentation Improvements:** See Weakness 2.
**Soundness:** 3

**Ethical Concerns:**

Yes

**Excitement:**

3: Ambivalent: It has merits (e.g., it reports state-of-the-art results, the idea is nice), but there are key weaknesses (e.g., it describes incremental work), and it can significantly benefit from another round of revision. However, I won't object to accepting it if my co-reviewers champion it.

**Justification For Ethical Concerns:**

Human evaluation is involved in this research. Authors need to confirm that an Institutional Review Board (IRB) review is completed prior to the study.

**Missing References:**

[1] MindCraft: Theory of Mind Modeling for Situated Dialogue in Collaborative Tasks. Cristian-Paul Bara et al. 2021

[2] How Well Do Large Language Models Perform on Faux Pas Tests? Natalie Shapira et al. 2023.

[3] Clever Hans or Neural Theory of Mind? Stress Testing Social Reasoning in Large Language Models. Natalie Shapira et al. 2023

**Paper Topic And Main Contributions:**

This paper evaluates large language models (LLMs) by introducing a language inference task of dynamic epistemic logic (DEL) problems. The problem formulation is cognitively motivated, and the focus is well-scoped under the knowledge and perception aspects of the theory of mind (ToM). The paper describes the techniques to create the MindGame datasets, and evaluates the performance of the Pythia and GPT families. The key finding of this work is that scaling does not guarantee better performance on DEL inferences, and that the task is challenging even for the best-performing LLMs like GPT-4.

**Questions For The Authors:**

Question A: Could the authors elaborate on the predicated used in Section 3.2, e.g., ONES, IDENTITY?

Question B: Could the authors elaborate on the experiment setups mentioned in Weakness 3?

Question C: I understand that the 4-page limit can be challenging for a short paper. If this paper gets accepted, what are the authors’ plans for the extra page?


**Reasons To Accept:**

Strength 1: The paper introduces the MindGames dataset. The empirical result shows that it is a challenging and useful testbed for ToM assessment in LLMs. There is a clear resource contribution for the commonsense reasoning and language modeling community.

Strength 2: The problem formulation is cognitively-motivated, and the problem definition in Section 3.1 is potentially generalizable to more complex social interactions.


**Reasons To Reject:**

Weakness 1: The paper needs to situate itself better in related work and be more cautious with the claims it made. I am worried about the authors' claim that this work “presents the first dataset with a complexity that can challenge supervised learning models (lines 65-66)”. First, to my knowledge, epistemic reasoning has been explored in terms of ToM in language models (as authors also cited in Section 2.1), and various other ToM benchmarks are known to be challenging (see missing references). Second, while authors argued that previous work adopts “human created standardized tests or rule-based templates” and “focus on simplistic reasoning and require further validation” (lines 6-9), I am not convinced that the task introduced by the authors is “complex” since it still relies on language templates and there is no analysis on potential shortcuts.

Weakness 2: This paper can benefit from an improvement in writing and presentation. For example, non-experts can get very confused about the exact meaning of “epistemic logic” as there seems to be no clear definition in this work. The setups in lines 61-63 need to be described in more detail and the predicates in Section 3.2 are never introduced. In this case, people without a psychology background can get confused. The example given in line 53 has too many pronouns and is hard to read even for human readers.

Weakness 3: The experiment setups need to be described in more detail. In section 4.3, only 20 sessions are randomly chosen for evaluation on GPT-4. Are the 20 sessions from the same setup as the performance on the 4 different setups clearly different? What is the hyper-parameter of inference using OpenAI API (e.g., temperature)? What are the probing techniques used (e.g., generating Y/N or perplexity)? Did the authors fix the context in 5-shot learning settings across different models?


**Reproducibility:**

3: Could reproduce the results with some difficulty. The settings of parameters are underspecified or subjectively determined; the training/evaluation data are not widely available.

**Reviewer Confidence:**

5: Positive that my evaluation is correct. I read the paper very carefully and I am very familiar with related work.

---

> ### Author Rebuttal · Authors · 2023-08-28
>
> We sincerely thank you for the thoughtful comments and valuable feedback on our submission.
>
> Weakness 1:
> The templates are bound to rules in the previous work we cited, while we used a form of context-free grammar, and a modal logic solver, which can involve diverse reasoning patterns. The complexity stems from the logic and not the templates. We will update our phrasing in the updated version.
> To control for shortcuts, we trained a relatively shallow supervised model (deberta-small, 6 layers, 44M backbone parameters) model on a training set combining all setups  (11.2k examples, 3 epochs, LR 3e-5). . Overall accuracy was 83%. Simpler lexical baselines did not capture negations well enough.
> We assumed that examples correctly predicted by deberta-small with high confidence contained shortcut cues.
> The dataset we enclosed with the initial submission already contains “deberta-pred” and “deberta-confidence” column. We found that LLM already failed on easy examples, so we did not eliminate them. Our dataset can be filtered by “difficulty” using deberta confidence level and the discrepancy between deberta-small prediction and ground truth.
>
> Weakness 2:
> We will split the examples of line 53 into multiple lines and replace pronouns by name using additional space, and define modal logic, epistemic logic, and dynamic epistemic more explicitly.
>
> Weakness 3:
> For the text completion evaluation:
> We used the default parameters of LM evaluation harness v0.3.0. (1.0 for Pythia, 0.0 for GPT-3, which is the default value for the completion API). As mentioned on line 214, we use perplexity to compare the score of true and false answers.
> Examples for few-shot learning are randomly selected from the same setup.
>
> For the ChatGPT experiments:
> We used the ChatGPT interface (starting new chats for each example) and the default OpenAI API for qualitative evaluation, using the prompt of the qualitative analysis examples (“Is the hypothesis true or false ?”).
>
>
> A: This notation represents special matrices (Identity(N) is the identity matrix of size N, Ones(N) is the all-ones matrix of size N×N).
>
> B: See answer to weakness 3.
>
> C: In addition to the requested references and clarifications, we would be glad to use the additional space to discuss the problem of shortcuts and detail the additional analysis derived from the automatic difficulty scoring of the examples.
>
> [1] Zhang, Honghua, et al. "On the paradox of learning to reason from data." arXiv preprint arXiv:2205.11502 (2022). https://arxiv.org/abs/2205.11502

---

### Official Review · Reviewer_GnXA · 2023-08-04

**Soundness:** 3

**Excitement:**

4: Strong: This paper deepens the understanding of some phenomenon or lowers the barriers to an existing research direction.

**Paper Topic And Main Contributions:**

This paper designed controlled experiments to assess PLMs' ability on Theory of Mind (ToM). The authors leveraged dynamic epistemic logic problems and verbalized these questions in English natural language. The results show that language model scaling does not consistently achieve better performance. The dataset contributes to the increasing debate on LMs' cognitive abilities, and to what extent scaling helps with the performance.

**Questions For The Authors:**

A. In the case of few-shot learning, was the type of examples systematically controlled? Does forehead-mud setup benefit or hinder the reasoning in foreheaf-mud-mirror setup? These two setups have similar structure, but different reasoning outcomes.

B. How does the use of negation in the first-order belief (can know that v.s. cannot know that) affect the performance.

**Reasons To Accept:**

1. This paper carefully designed a fairly challenging dataset based on dynamic epistemic logic problem to understand Theory of Mind. The dataset is systematically controlled with different variations, and a template for verbalization.

2. They conducted scaling experiment with the designed dataset, showing that only scaling only improves 5-shot but not 0-shot reasoning. This raises the question whether language models' emergent behavior of reasoning demonstrate their ToM capacity or simply imitating the pattern.

**Reasons To Reject:**

1. [Human evaluation] The dataset is only evaluated by two NLP researchers instructed to perform entailment detection. Given this task is fairly challenging for humans when the situation only communicated verbally, I wonder if the verbal set-up has increased unnecessary difficulty for language models. It would be helpful to see baseline human evaluations, to disentangle the challenge to imagine a situation based on languages, and to perform ToM and perspective taking.

2. [Variety of set-ups] The authors have four set-ups (forehead-mud, forehead-mud-mirror, thirst, explicit), but did not provide any detailed by-type analysis on these variations. These details would be important to include to understand the nature of ToM, as these set-ups do not have similar complexity. For example, the forehead-mud-mirror setup requires the understanding the consequence of looking at the mirror in addition to forehead-mud setup.

3. [Statistical association] While the verbalization provides a systematic way to scale up and generate test cases, it hinders the naturalness of language and make it difficult to tease apart reasoning from mimicking the statistical patterns, especially in the case of few-shot reasoning. The dataset would benefit by introducing more varieties of language template other than "it is publicly announced that someone", with detailed analysis on how the linguistic properties of the template is affecting the results. For instance, the recency of mentioning the target and the use of discourse connections would likely to change the reasoning accuracy.

**Reproducibility:**

4: Could mostly reproduce the results, but there may be some variation because of sample variance or minor variations in their interpretation of the protocol or method.

**Reviewer Confidence:**

3: Pretty sure, but there's a chance I missed something. Although I have a good feel for this area in general, I did not carefully check the paper's details, e.g., the math, experimental design, or novelty.

---

> ### Author Rebuttal · Authors · 2023-08-28
>
> We sincerely thank you for the thoughtful comments and valuable feedback on our submission.
>
> [Human evaluation]
> We could disantangle language from reasoning by showing visual scenarios to humans, but human accuracy is already relatively high, and the objective our our study is to test verbal scenarios on LLM. Human evaluation acts as a sanity check, to validate the verbalization of the logic problems and the adequation of the logic solver.
>
>
> [Variety of setups]  The setups require similar types of reasoning (an agent knows something if it is true in all possible situations).
> The mirror setup is not necessarily more complex, as it mostly acts as a shortcut (an agent can look at itself and knows that other agents can look at themselves).
> The main difference lies in the commonsense assumptions about what information is initially accessible in specific situations. The explicit setup ablates that aspect by stating explicitly what is initially accessible. Our results suggest that this does not help.
>
> [Statistical association]
> We chose explicit language over fluency, but ensured that the text is human readable. The reasoning is the center point of our work.
> We note that there also are statistical associations in the logic itself [1].
> For instance, specific claims have less chance to be true. (“someone knows that someone is muddy” is more likely that “Mary knows that Paul is muddy”, regardless of the premise). To control for this type of phenomena, we trained a relatively shallow supervised model (deberta-small, 6 layers, 44M backbone parameters) model on a training set combining all setups (11.2k examples, 3 epochs, LR 3e-5). Overall accuracy was 83%. Simpler lexical baselines did not capture negations well enough.
> We assumed that examples correctly predicted by deberta-small with high confidence contained cues.
> The dataset we enclosed with the initial submission already contains “deberta-pred” and “deberta-confidence” columns. We found that LLM already failed on easy examples, so we did not eliminate them. Our dataset can be filtered by “difficulty” using deberta confidence level and the discrepancy between deberta-small prediction and ground truth.
>
>
> A: We controlled the type of examples in the few-shot learning, to be charitable to the language models.
>
> B: The performance is low in both cases (when comparing accuracy when the hypothesis contains vs. does not contain negation).
>
> [1] Zhang, Honghua, et al. "On the paradox of learning to reason from data." arXiv preprint arXiv:2205.11502 (2022). https://arxiv.org/abs/2205.11502

---

### Meta-Review · Area_Chair_TkEV · 2023-09-19

**Recommendation:** 4

**Metareview:**

The paper introduces a challenging dataset to assess LMs' abilities in Theory of Mind (ToM). The reviewers find the paper well-written and clear, and they consider the approach novel, sound, and interesting. The authors have successfully addressed the concerns raised by the reviewers. I recommend accepting the paper.

---

### Decision · Program_Chairs · 2023-10-07

**Decision:**

Accept-Findings

**Comment:**

The paper introduces a challenging dataset to assess LMs' abilities in Theory of Mind (ToM). The reviewers find the paper well-written and clear, and they consider the approach novel, sound, and interesting. The authors have successfully addressed the concerns raised by the reviewers. I recommend accepting the paper.